# A Perspective on Current Flavivirus Vaccine Development: A Brief Review

**DOI:** 10.3390/v15040860

**Published:** 2023-03-28

**Authors:** Sudip Kumar Dutta, Thomas Langenburg

**Affiliations:** Artemis Bioservices, Molengraaffsingel 10, 2629 JD Delft, The Netherlands

**Keywords:** flavivirus, vaccines, dengue, Zika, West Nile, yellow fever, Japanese encephalitis and tick-borne encephalitis, clinical trials

## Abstract

The flavivirus genus contains several clinically important pathogens that account for tremendous global suffering. Primarily transmitted by mosquitos or ticks, these viruses can cause severe and potentially fatal diseases ranging from hemorrhagic fevers to encephalitis. The extensive global burden is predominantly caused by six flaviviruses: dengue, Zika, West Nile, yellow fever, Japanese encephalitis and tick-borne encephalitis. Several vaccines have been developed, and many more are currently being tested in clinical trials. However, flavivirus vaccine development is still confronted with many shortcomings and challenges. With the use of the existing literature, we have studied these hurdles as well as the signs of progress made in flavivirus vaccinology in the context of future development strategies. Moreover, all current licensed and phase-trial flavivirus vaccines have been gathered and discussed based on their vaccine type. Furthermore, potentially relevant vaccine types without any candidates in clinical testing are explored in this review as well. Over the past decades, several modern vaccine types have expanded the field of vaccinology, potentially providing alternative solutions for flavivirus vaccines. These vaccine types offer different development strategies as opposed to traditional vaccines. The included vaccine types were live-attenuated, inactivated, subunit, VLPs, viral vector-based, epitope-based, DNA and mRNA vaccines. Each vaccine type offers different advantages, some more suitable for flaviviruses than others. Additional studies are needed to overcome the barriers currently faced by flavivirus vaccine development, but many potential solutions are currently being explored.

## 1. Introduction

The genus flavivirus consists of around 70 species, of which 30 are arthropod-borne viruses that can cause disease in humans (Figure 1). Six of these viruses are considered medically relevant, namely dengue (DENV), Zika (ZIKV), yellow fever (YFV), tick-borne encephalitis (TBEV), Japanese encephalitis (JEV) and West Nile (WNV), which are transmitted through mosquitos and ticks [1,2]. Together, these six flaviviruses cause an immense global burden, with billions of people in the endemic regions (Table 1). Over the last decades, the geographical distribution of the flavivirus vectors has expanded due to climate change, which contributes to increasing number of flavivirus infection.

### 1.1. Epidemiology of Flaviviruses

The global yearly incidence of DENV infections alone is 5%, with an annual estimate of 500,000 hospitalizations and 20,000 deaths [4,5]. DENV is transmitted through bites of *Aedes* (*Ae*) *aegypti* or *Ae. albopictus* mosquitos, whose intercontinental distribution is growing through climate change. There are four distinct DENV serotypes, and while there are some differences in nucleotide and amino acid levels, they cause similar symptomatology [6]. Primary infection of DENV will mostly cause mild disease and confer life-long immunity for that specific serotype. Secondary infection with a different DENV serotype has a higher risk of severe disease due to presumably antibody-dependent enhancement (ADE) [7]. About 1 out of 4 people infected with DENV are symptomatic, whilst approximately 1 in 800 people become hospitalized with severe dengue. Patients can experience mild aspecific symptoms such as fever, rash and vomiting, whereas severe infections can cause a syndrome called hemorrhagic fever, characterized by hematemesis and fluid accumulation in the thorax or peritoneum, eventually resulting in shock syndrome (Table 1) if fluid replacement therapy is not started immediately. DENV, as well as ZIKV, are prevalent in Latin America, Africa, Asia and the Pacific Islands and are transmitted by the same mosquito species, which makes several regions endemic for both viruses (Figure 2 and Figure 3) [8,9]. Presently, 89 countries have reported evidence of occurred ZIKV transmission, although data on case incidence and hospitalizations are limited [10]. Most transmissions occur through mosquito bites, while transmission through sex is possible. ZIKV infections are mostly asymptomatic or cause mild symptoms such as fever, rash and headaches. The greatest burden is caused by ZIKV infection during pregnancies. It has been demonstrated that ZIKV can penetrate the placental barrier and spread illnesses to the developing fetus [8]. This causes infants to be born with birth defects such as microcephaly or other congenital malformations.

YFV is transmitted in an urban or sylvatic cycle by *Aedes* or *Haemogogus spp*. mosquitos in the tropical and subtropical regions of Latin America and Africa [9]. An estimated 200,000 cases of YFV are reported annually, of which 30,000 result in death, with severe cases and deaths predominantly occurring in Africa [14]. After an incubation period, mild symptoms appear such as fever, nausea, myalgia and dizziness. In 20% of infected people, the disease progresses to a more severe stage, which is characterized by high fever, jaundice and bleeding. Until the early 1990s YF was considered a major threat, and massive vaccination campaigns were conducted in the endemic regions, resulting in a decrease in the number of cases. Thus, by the end of the 20th century, the status of YF has been changed to neglected tropical disease [19]. However, the reemergence of major YF infections has been reported during 2008–2009, 2010–2015 and 2017–2018 from endemic and nonendemic areas of South America during the past decade [20]. After analyzing the outbreak samples, Haslwanter et al. indicated that the 2017–2019 YFV epidemic in Brazil was associated with an emerging strain, YFV 2017–2019. Based on a recent report published by WHO in 2022, approximately four million people have been vaccinated in Africa, with effective vaccines developed against YFV by early 2021. However, 183 confirmed cases and 21 deaths from that region indicate ongoing virus transmission.

JEV is endemic in 25 countries in Asia and the Pacific Islands, where it is mainly transmitted through *Culex tritaeniorhynchus* [21]. The symptomatic disease occurs at varying rates, depending on age and geographic region. For instance, the incidence is lower in people living in endemic areas (1:300) and higher in people from non-endemic countries (1:25). Those with mild disease exhibit symptoms such as fever, headache and vomiting [15]. These symptoms may progress to severe headaches, paralysis or seizures in the form of acute encephalitis syndrome (AES), which has a high mortality. Patients who survive AES may suffer from long-term neurologic, cognitive and/or psychological effects. A study conducted by Caldwell et al. in 2018 estimates a speculative global incidence of 66,000 annual cases [22]. On 19th October 2022, a report was issued by the Australian government based on 42 confirmed human cases of JEV from 5 states, indicating the movement of the virus from the southern to the mainland region [16]. WNV is transmitted by *Culex spp* as well, in particular, *Culex pipiens*., found in Africa, Europe, the Middle East, North America and west Asia [12].

Approximately 20% of WNV infections are symptomatic, which start with fever, headache and body aches, whereas 1 in 150 WNV patients experience severe and potentially fatal disease [13]. Severe WNV infections result in meningitis and/or encephalitis, with symptoms such as high fever, myalgia, numbness and, in some cases, paralysis, vision loss and coma can occur. Interestingly, 863 confirmed WNV cases and 58 deaths were reported from multiple parts of the United States of America and 962 WNV infections and 72 deaths were documented from different parts of the European Union (EU) and European Economic Area (EEA) due to WNV infection [23]. TBEV is mostly transmitted by *Ixodidae ricinus* and is endemic in Europe, Russia, north China and Japan, with around 10,000 to 12,000 annually reported cases [17]. Similar to the other medically relevant flaviviruses, initial symptoms consist of fever, headache, myalgia, fatigue, meningitis, encephalitis and myelitis. The symptomatic disease remains a febrile illness in some cases, but generally progresses to meningitis, meningoencephalitis or meningoencephalomyelitis, causing a variety of neurologic symptoms. Radzišauskienė (2020) reports that between 2005 and 2017, 14.6% of TBEV cases in Lithuania led to severe disease [18]. According to the annual report published by the European Center for Disease Prevention and Control, multiple confirmed cases (*n* = 3734) of TBEV from 24 EU and EEA countries were recorded until the end of October 2021 [24].

### 1.2. Molecular Characterization and Life Cycle

Flaviviruses are positive-sense, single-stranded RNA viruses that share a common genomic organization and life cycle and several host–pathogen protein interactions [25]. Similar to most enveloped virions, they are composed of the nucleocapsid (C), which protects the single-stranded RNA genome, surrounded by a lipid bilayer that contains the membrane (M) and envelope (E) proteins [26]. The RNA genome ranges from 10 to 11 kb, which encodes a 3′ and 5′ untranslated region and a long open reading frame (ORF), which translates into a large polyprotein. This polyprotein is processed by co- and post-translational modification by viral and host proteases to form three structural proteins and seven non-structural proteins. The non-structural proteins, namely NS1, NS2A, NS2B, NS3, NS4A, NS4B and NS5, are involved in viral RNA replication, modulation of the host response and viral assembly.

Together with co-factors NS2B or NS4A, NS3 is a multifunctional protein that serves as a helicase, has nucleoside tri-phosphatase activity and works as the primary viral protease. NS5 forms the viral RNA-dependent RNA polymerase and is involved in immune evasion and modulation. NS1 plays a role in the early phase of replication, whereas NS2A is involved in viral assembly. NS2A, NS4A, NS4B and NS5 together form the replication complex, and NS1, NS2A, NS4B and NS5 are all involved in the evasion or modulation of the host response. Viral entry occurs by receptor-mediated endocytosis, after which the virions travel to the endosomes [27]. The acidic environment of the endosomes results in conformational changes in the E protein, which induces the fusion of host and viral membranes. The RNA genome is released, and the polyprotein of around 3400 amino acids is transcribed. Following translation and modification of the polyprotein, NS5 synthesizes the minus RNA strand from the RNA genome, which serves as a template for the positive strand. Replication occurs on virus-induced host membranes, which serve as scaffolding for the replication complex. Immature non-infectious viral particles are assembled in the endoplasmic reticulum (ER). These particles are transported through the host secretory pathway to the trans-Golgi network, where maturation occurs. Subsequently, the mature particles are released from the cell through exocytosis [28].

### 1.3. Immune Response

Production of neutralizing antibodies (nAbs) is crucial in the immune response against flaviviruses. Rey et al. reported that the main target of the antibody response against flaviviruses is the E protein [29], however, the M and NS1 proteins have also been reported to be targeted by monoclonal antibodies [30]. E protein located on the virus surface consists of three domains (demarcated as EDI, EDII and EDIII), and antibodies recognizing and binding with domain I and II are proposed to produce more potent neutralizing activity than antibodies interacting with domain III. However, in the case of serotype-specific antigen-antibody response, domain III-recognized antibodies were reported to produce nAbs that are more potent than those from domain I and II. A study conducted on human subjects infected with DENV by de Alwis et al. in 2012 reported that a lower level of EDIII-recognizing antibodies was identified in humans [31]. The mature viral particle is highly dynamic due to the E protein undergoing constant conformational changes. This results in a variation in the accessibility of specific epitopes, which, in turn, hinders the antibody recognition. The precursor membrane protein (prM), important during virus assembly, was reported to enhance the accessibility of the hydrophobic fusion loop located in EDI and EDII, facilitating the production of the higher level of nAbs. Thus, in absence of prM, antibodies targeting epitopes located on the fusion loop show a lower level of neutralization and a high level of ADE because of their high cross-reactivity nature among flaviviruses. Structural heterogeneity can also occur in the M protein due to inefficient cleavage, resulting in a wide variation of partially mature and fully mature viral particles [30]. These components with a high variety of accessible epitopes might promote ADE in flavivirus infections. Interestingly, a higher level of protection and no ADE were reported among a wide range of flaviviruses in vitro (viz. DENV, ZIKV and WNV), mediated by virus-specific NS1 monoclonal antibodies (mAbs) [32]. Falconar, in 1997, reported the potential pathogenic role of antiNS1 antibodies due to their cross-reactive nature with human endothelial cell monolayers and thrombocytes [33]. However, Batty et al. in 2015 demarcated the protective nature of antiNS1 antibodies on endothelial cells [34]. CD4+ and CD8+ T-cells also play a crucial role in the immune response against flaviviruses. Campos et al. [30] report that the amplitude of T-cells plays a role in the severity of the disease. Cell-mediated cytotoxicity is needed for the early containment of the pathogen. However, excessive T-cell infiltration and production of proinflammatory cytokines might lead to damage to organs and increased disease severity. Due to the high sequence identity between flaviviruses, a secondary infection might lead to cross-reactive T-cell or nAbs responses. Memory cells generated during the primary infection might overflow and compromise a secondary response. Severe dengue observed in secondary DENV infections is associated with ADE, although the exact immunopathology of DENV still remains controversial [34].

### 1.4. Flavivirus Vaccines

Prevention and control strategies through vector control have been beneficial in reducing the infection rate and burden of disease, although these strategies face substantial limitations that inhibit their efficacy [35]. Moreover, to this day, no specific antiviral treatment has been developed for flaviviruses. Therefore, due to its cost effectiveness and potential for developing long-lasting immunity, vaccination (which may be defined as a biological preparation that gives active acquired immunity against a specific illness) is the most alluring strategy for reducing the worldwide burden of flaviviruses [36]. An ideal vaccine should be safe (even in individuals with a weak immune system), preferably provide sterile protection against the infection and, upon administration, rapidly activate B-cells, resulting in the production of an adequate level of protecting/neutralizing antibodies, as well as activation of T- (helper, memory and cytotoxic) cells. CD4+ T-cells produced after vaccination are identified as a key factor responsible for the production of specific neutralizing antibodies by B-cells, and CD8+ T-cells identify the infected cells by recognizing the peptide presented by major histocompatibility complex (MHC) and kill them [37,38]. Due to the short lifespan of B-cells in blood, with due time, the antibody titer is reduced, resulting in compromising long-term protection against pathogens. Therefore, to obtain long-term protection, the immunogen must trigger the B-cell germinal center, thus producing a subpopulation of B-cells known to trigger a robust antibody-mediated secondary immune response during re-encounter with wild-type and/or mutated virus [39,40]. Different complementary immune responses are induced by vaccines to achieve long-term protection against different virus subtypes. Moreover, the vaccine must be economical, stable during transport and easily administrated. However, vaccine development for flaviviruses faces challenges and limitations as well. The complicated immunopathology of certain flaviviruses and the absence of ideal research tools have inhibited the development of efficient and safe vaccines [41]. Vaccines can be broadly classified into live-attenuated, inactivated, recombinant and genomic vaccines (Figure 4). Live-attenuated vaccines utilize a version of the living pathogen which has been attenuated in a way it does not cause disease in healthy individuals. Due to the similarity with a natural infection, live-attenuated vaccines provide strong immunity, often only needing a single immunization. However, there is the possibility that the weakened pathogen reverts to its disease-causing state, which is why it cannot be administered to immunocompromised people [36]. That issue is non-existent with inactivated vaccines, for which a method is used to inactivate the pathogen by heat, formalin or formaldehyde, leaving the pathogen intact but unable to replicate. Inactivated vaccines do need several doses, as they require boosters and, in some cases, also the use of an adjuvant to induce long-lasting immunity. Likewise, recombinant vaccines and genomic (DNA or RNA) vaccines require booster doses, although these types only use parts of the pathogen [42]. Using only components of a pathogen often increases safety but reduces immunogenicity. The different recombinant vaccine types that will be discussed are subunit, VLPs, viral vector-based and epitope-based vaccines. Additionally, genomic vaccines based on plasmid DNA and mRNA will be included as well. Currently, FDA-approved live-attenuated and inactivated vaccines are available for YFV, JEV and TBEV, whereas various vaccines for DENV, ZIKV and WNV are in different phase trials [43]. This review is mainly focused on the successes, as well as the challenges that have been faced by flavivirus vaccine development to gain insight into research strategies for potential flavivirus vaccines.

### 1.5. Vaccine Development

The development of vaccines consists of a pre-clinical stage and a clinical stage [44]. Preclinical testing includes all aspects of testing prior to testing in humans and aims to provide in vitro and in vivo evidence regarding proof of concept and safety [45]. This includes product characterization, proof of concept/immunogenicity studies and safety testing in animals. According to the WHO guidelines on the non-clinical evaluation of vaccines, these aspects are prerequisites before starting clinical trials [46]. Product characterization consists of identifying and assessing important elements for the design of vaccines, such as the starting materials, manufacturing process and test methods. The quality, safety and potency of vaccine products are sensitive to changes in the manufacturing process, which is why consistency in the production is essential. Subsequently, potency tests are performed to verify the consistency of the manufacturing process. Furthermore, vaccine batches tested in the preclinical stage should be adequately representable as the batches used in clinical testing to ensure reliability. Immunogenicity studies are pharmacodynamic studies conducted to assess and evaluate the relevant immune responses induced in vaccinated animals. These studies are conducted in animal models, as they provide “proof of concept” information to support a clinical development plan. Additionally, immunogenicity studies assist in determining immunological characteristics and in ratifying the doses, schedules and routes of administration for clinical trials. The response of each antigen within a vaccine should be assessed. Safety testing in animals is essential for identifying potential safety concerns for clinical testing such as the potential inherent toxicity of the product or toxic side effects as a result of the preparation or the immune response. Study designs need to take several parameters into account, such as relevant animal species and strain, dosing schedule, method of vaccine administration and timing of endpoints. The methods of testing should correspond to the intended strategies in clinical testing where it is feasible. These non-clinical studies continue during the clinical stage if changes in the product manufacturing occur, or to further study safety concerns arising from the clinical stage. 

Following the analysis and evaluation of non-clinical testing, a vaccine candidate might progress to clinical trials. The WHO guidelines for good clinical practice describe phases in which clinical trials need to be performed [47]. First, preliminary trials are conducted to assess the human immune responses to the antigenic components, also referred to as Phase I trials. Generally, Phase I trials involve 20–100 healthy adults, preferably subjects with no prior exposure to the target pathogen [48]. Secondly, additional clinical trials are conducted to further assess the immunogenicity and drug safety: the Phase II trials. Phase II trials include several hundred subjects who, if feasible, represent the target population intending to be treated. Phase I and II trials are usually designed in order to provide sufficient immunogenicity and safety data for evaluation in Phase III trials. Phase III trials, or “pivotal trials”, are conducted to provide clinical evidence for the licensure of the vaccine candidate [49]. Usually involving thousands of participants, Phase III trials can take several years. Several criteria might be evaluated in Phase III trials in order to determine the efficacy of the vaccines, to provide an indication of the ability of the vaccine to protect by using immunogenicity data or to assess specific safety concerns. This concludes a simplified structure of vaccine development, the methods used in the actual development vary between vaccine types and products.

### 1.6. Live-Attenuated Vaccines

Live-attenuated vaccines are composed of a live pathogen that has been attenuated in a way that it does not cause serious disease [50]. Live-attenuated vaccines can be designed to induce an immune response to themselves or a heterologous antigen [51]. Several effective and successful live-attenuated vaccines have been developed for preventing diseases such as smallpox and polio, as well as the flavivirus vaccines against YFV and JEV [52,53]. Live-attenuated vaccines mimic natural infections and, as a result, they can induce strong immune responses and confer immunity that lasts for decades, even with only one dose. However, due to the use of a live strain of a microorganism, the possibility exists that it might revert to a pathogenic state [51]. Therefore, the goal of developing live-attenuated vaccines is to attenuate the microorganism to be as avirulent as possible while still upholding enough proliferation to induce a sufficient immune response and conferred immunity. The classical method of attenuation was through continuous passage and selection of a candidate strain; however, this would sometimes lead to the strain reverting to a pathogenic state. Subsequently, several rational methods that further reduce the probability of a reversion to a pathogenic state have been developed over the years [54]. The method of attenuation by loss of genetic pool has been used for the development of YFV, measles and influenza vaccines. Due to the rapid mutations in viral species, viral populations do not consist of a single genotype, but rather a mix of genotypes. This abundance of genetic diversity is an essential virulence factor, as it allows the viral pathogen to quickly adapt to the host environment. The genotypic diversity can be dismantled by propagating the candidate strain in an atrophic host, which subjects it to continuous variation and competition. By restricting the genotypic diversity, the virulence of a virus decreases. Another approach to attenuation is targeting and deoptimizing codons [55]. Since the expression of synonymous codons (multiple codons can code for a single amino acid) can differ per organism, the frequency of those codons might be unequal, referred to as a codon usage bias. Optimization of codon usage is a prerequisite for efficient gene expression, therefore deoptimization of codon usage may lead to reduced expression of essential viral proteins. By altering the codon composition, the virulence of the pathogen can be greatly reduced. However, the proteins expressed are still identical, thus eliciting immune responses toward natural viral proteins. This approach is applicable to a large number of viruses, and, by deoptimizing thousands of synonymous codons with point mutations, the probability of a reversion to a pathogenic state becomes minimal. Several other attenuation approaches exist, such as microRNA-mediated attenuation, inducing targeted mutation in essential proteins or modification of cleavage sites [56]. 

As mentioned earlier, live-attenuated vaccines have already been successful against some flaviviruses, namely YFV and JEV (Table 2). In the 1930s, the 17D live-attenuated vaccine was developed against YFV [57]. The 17D vaccine was attenuated by passaging the wild-type Asibi isolated from a human patient in west Africa through mouse and chicken embryos. Three substrains of the 17D vaccine are presently in production: the 17DD (Brazil), 17D-213 (Russia) and 17–204 (China, France, Senegal and the USA) vaccines. The 17D vaccine confers immunity in 95% of recipients in a single dose. While the vaccine is safe for most recipients, there have been reports of rare complications described as vaccine-associated viscerotropic disease (YEL-AVD) and vaccine-associated neurotropic disease (YEL-AND). Although the incidence of YEL-AVD is extremely low (2.5 per million), it is still higher than severe or life-threatening events associated with other licensed vaccines [58]. It is important to note that the 17D vaccine provides long-lasting immunity against severe disease, however, safer alternatives are being considered. For JEV, two live-attenuated vaccines have been developed: the SA14-14-2 JE vaccine and the IMOJEV vaccine [59,60]. Derived from the S14 wild-type strain, the SA14-14-2 JE vaccine was developed by passaging the wild-type strain in hamster kidney cells. While inducing strong and long-lasting cross-reactive protection, the vaccine shows an immunization rate of 85–100% without any major vaccine-induced side effects. The IMOJEV vaccine is a chimeric live-attenuated vaccine wherein the previously discussed YFV-17D vaccine is used as a vector [41]. Multiple chimeric live-attenuated vaccines have been developed with 17D as a vector using the ChimeriVax technology platform. For IMOJEV, the ChimeriVax platform was used to replace the cDNA encoding the E and M proteins with the respective proteins of the SA14-14-2 JE vaccine by recombinant DNA technology. The use of the YFV-17D vaccine as a vector comes with the risk of YEL-AVD, however, studies suggest that the E and M proteins of YFV play a role in viscerotropic, and the substitution of those proteins may affect this mechanism. Another licensed chimeric live-attenuated vaccine is Dengvaxia (CYD), although it is limited to only two countries, Brazil and the Philippines, where the vaccine uptake is low [61]. The reasons for this are the increased risk of severe dengue and the high costs. Dengvaxia uses the YFV-17D vaccine as a vector as well by combining genes from E and M proteins of the four DENV serotypes with the non-structural genes from YFV-17D through recombinant DNA technology. Dengvaxia is tetravalent, consisting of four monovalent chimeric vaccine viruses: CYD-1, CYD-2, CYD-3 and CYD4. Dengvaxia has shown variating efficacy between DENV serotypes and even poor efficacy in DENV1 and DENV2. Henein et al., in 2017, showed that the elicited nAbs are mostly specific to only DENV4 and reported that this is likely due to interference among the monovalent vaccine viruses [62]. However, the low antibody titers did not seem to correspond to low efficacy, therefore another issue with Dengvaxia is the absence of reliable surrogate markers for vaccine efficacy [63]. Moreover, long-term safety studies showed an increase in severe dengue in young children. Additional safety studies were performed, which showed that seronegative recipients of all ages had increased rates of severe dengue [64]. These results strongly suggest that Dengvaxia makes seronegative recipients more susceptible to ADE. According to Thomas et al., 2019, a reason for this might be that Dengvaxia mimics a primary infection in seronegative, increasing the risk for severe dengue in the case of secondary infection [63]. This theory assumes that Dengvaxia does not confer immunity for one or more DENV serotypes. Currently, the WHO advises only administering Dengvaxia to seropositive recipients.

Presently, there are several chimeric live-attenuated vaccines in clinical trials for both DENV and WNV (Appendix A). Tetravax-DV is a chimeric live-attenuated DENV vaccine in Phase II and III trials. Tetravax-DV consists of 4 monovalent DENVs attenuated by targeted 30-nucleotide deletions in a 3′NTR genome region [65]. In addition, the Takeda vaccine, which is based on four chimeric monovalent DENVs as well, is attenuated through passaging a DENV-2 strain in primary dog kidney cells (PDK) [66]. Phase II and III trials are currently ongoing, with previous Phase I/II trials showing promising results regarding immunogenicity and safety in both adults and children. A third DENV chimeric live-attenuated vaccine is the CYD-TDV vaccine, which was developed by passaging four monovalent strains in PDK and fetal Rhesus cells [67]. Phase II trials showed adequate immunogenicity and safety, however, at long-term follow-up, it became clear that humoral immunity was not durable. Two chimeric live-attenuated vaccines for WNV are currently in clinical trials, and both have completed Phase I trials. WN/DEN4∆30 utilizes a DENV backbone with the WNV E and M proteins, as well as 30-nucleotide deletions in the 3′UTR region of the DENV-4 non-structural genes [68,69]. Three Phase I trials were conducted, which showed satisfactory efficacy and tolerance, making WN/DEN4∆30 a suitable candidate for Phase II trials. The second WNV chimeric live-attenuated vaccine is ChimeriVax-WN02, a chimeric vaccine consisting of the WNV NY99 strain with the YFV 17D non-structural genes with enhanced attenuation through point mutations in the E protein [69,70]. Two Phase II trials have been conducted, which showed adequate immunogenicity and safety as well as the highest seroconversion rates compared to the other six vaccine candidates for WNV.

### 1.7. Inactivated Vaccines

As the name clearly indicates, that inactivated vaccines involve approaches of first inactivating or killing the pathogen. Viral particles are grown in cell cultures, after which they can be inactivated or killed through heat inactivation or radiation or with chemicals such as formalin [71]. Consequently, the virulence of the pathogen is greatly reduced, potentially eliminating the probability of infection. The inactivation procedure will result in the viral particles either being destroyed or disrupted. Nevertheless, the viral capsid proteins should remain intact and recognizable by the immune system. The immune response to inactivated vaccines is mostly humoral and relies on the production of neutralizing (nAbs), with little to no cellular immunity being induced. The formulation of inactivated vaccines may include adjuvants or human serum albumin to stabilize the virus. In comparison to live-attenuated vaccines, inactivated vaccines provide higher safety than live-attenuated vaccines due to the inability of the viral particles to establish an infection. Therefore, inactivated vaccines are administrable to infants and immunodeficient people. However, since there was no infection, the initial dose did not result in immunity, therefore booster doses were necessary. Moreover, if the inactivation procedures are not adequately performed, the probability exists that infectious viral particles might be administered [72]. The inactivation may also lead to damage of viral-neutralizing epitopes, resulting in insufficient nAb levels. Therefore, during the development of inactivated vaccines, it is essential to conduct efficient quality control to assess the vaccine’s properties. 

Inactivated and live-attenuated vaccines are both classified as first-generation vaccines and together they make up all the currently licensed flavivirus vaccines. Presently, inactivated vaccines for JEV and TBEV are commercially available (Table 2), with several inactivated vaccines in phase trials for DENV, ZIKV, WNV and YFV (Appendix A). For JEV, two inactivated vaccines have been developed, JE-VC and JE-MB [73]. JE-VC is a Vero cell culture vaccine derived from the SA14-14-2 strain, whereas JE-MB is a mouse brain-derived vaccine from the JEV Nakayama strain. Both vaccines showed high and similar efficacy, and, for a long time, the JE-MB vaccine was the most frequently used JEV vaccine. However, Lindsey et al. in 2010 reported that the JE-MB vaccine has higher risks of hypersensitivity and neurological adverse reactions, and JE-VC was regarded as the safer alternative [74]. Five licensed vaccines have been developed for TBEV, which are all vaccines inactivated by formalin [75]. They have all been commercially available for decades, with some adjustments throughout the years. Although some of the TBEV vaccines’ licensures are restricted to only countries such as Russia, China or India, most of them share similar efficacy and safety profiles. Development strategies mostly differ between the selected strains. Encepur and FSME-IMMUN are almost identical in the production process, both cultivated from the same cell culture and stabilized with human serum albumin [76]. The KFD vaccine is derived from chicken embryo fibroblasts and shows reasonable efficacy but needs annual booster doses. TBE-Moscow and EnceVir are both based on the TBEV-FE strain and show similar efficacy and safety as Encepur and FSME-IMMUN. However, EnceVir does show an increased risk of fever and allergic reaction and is not used in children [77]. 

Presently, there are eight inactivated vaccines being tested as candidates for flavivirus vaccines in phase trials. We will not discuss every single candidate, but we will briefly discuss the relevance. One tetravalent inactivated vaccine is being developed for DENV, called TDENV-PIV, which showed promising results in a Phase I trial [78]. This candidate uses a formulation of four DENV strains (West Pac 74 (DENV-1), S16803 (DENV-2), CH53489 (DENV-3) and TVP360 (DENV-4), which were propagated in Vero cells and inactivated by formalin. With aluminum hydroxide as an adjuvant, TDENV-PIV showed sufficient immunogenicity and no adverse advents and is now heading for Phase II trials. Four inactivated vaccines are currently active in or have completed Phase I trials, which were all shown to be well-tolerated and elicit nAbs. Multiple purified inactivated ZIKV vaccine candidates have been entered into Phase I trials, out of which four have completed the Phase I trial. Based on available data, two out of four vaccines derived from the same ZIKA strain (PRVABC59 strain), namely ZPIV (purified, formalin-inactivated Zika vaccine) and PIZV (purified, formalin-inactivated, alum-adjuvated whole Zika virus vaccine candidate) were reported to be safe and well tolerated and induced substantial amounts of antibodies in healthy individuals [79,80,81]. For WNV, a hydrogen peroxide-inactivated vaccine has been developed, which did not show strong immunogenicity in Phase I trials but is currently being improved. In addition, a formalin-inactivated vaccine for WNV has been tested in Phase II trials, however, limited data are available on the immunogenicity. Additionally, an inactivated vaccine is in phase trials for YFV [82]. Since YFV vaccines bring the potential risk of yellow fever vaccine-associated viscerotropic disease (YEL-AVD) and other severe complications, a B-propiolactone inactivated YFV vaccine based on the 17D strain is being tested in clinical trials [83]. This candidate, XRX-001, induced strong immunogenicity and safety and may be a strong candidate to be a safer alternative for YFV vaccines.

### 1.8. Recombinant Vaccines

All commercially available vaccines for flaviviruses are exclusively live-attenuated or inactivated vaccines. These types have been successful against YFV, JEV and TBEV, although both types have encountered some limitations. At this moment, numerous newer generation vaccines such as subunit, virus-like particles and genomic vaccines are currently in development [36,41]. These types bring several advantages to the table, such as improved safety, although sometimes at the cost of efficiency. Whereas live-attenuated and inactivated vaccines contain the whole organism, recombinant vaccines induce immune responses using only parts of or parts similar to those of the virus.

#### 1.8.1. Subunit Vaccines

Subunit vaccines can involve components such as microbial proteins, synthetic peptides and carbohydrate antigens. By only using the parts of an organism that induce a desired immune response, subunit vaccines can target specific epitopes without the presence of potential pathogenic components. Accordingly, subunit vaccines produce highly safe and consistent immune responses [84]. However, since subunit vaccines do not have the capacity to establish an infection, the potency of the induced immune response is weaker than those of live-attenuated vaccines. Therefore, multiple booster doses are needed at several points in the patient’s life in order to provide long-lasting immunity. Moreover, most subunits vaccines need to include adjuvants, often aluminum salts, in order to elicit a stronger immune response. This adds additional safety testing, as the safety of adjuvants needs to be assessed both with and without the subunit vaccine components. Several other components may be added to the mixture, such as delivery systems or targeting moieties, in order to bring the vaccine in contact with specific immune cells. 

No subunit vaccines for flaviviruses are commercially available, although three subunit vaccines are currently in clinical trials (Appendix A). The tetravalent V180 vaccine for DENV utilizes a recombinant truncated polyprotein that holds 80% of the N-terminal DENV E protein as well as ISCOMATRIX™ as an adjuvant. This polyprotein is produced using the *Drosophila melanogaster* S2 expression system. Preclinical studies showed promising results, eliciting high nAbs titers in animal studies. Several formulations of V180 were tested in Phase I trials, which showed that all formulations with ISCOMATRIX™ showed robust immunogenicity [85]. However, more adverse events were observed in the formulations with ISCOMATRIX™ compared to the aluminum-adjuvanted or unadjuvanted formulations. Moreover, the DENV V180 vaccine, which comprises the recombinant DEN-80E envelope glycoprotein for each serotype, showed lower nAbs levels and durability for DENV4. Manoff et al. in 2019 also reported that this might be due to the short vaccination schedule, and that preclinical data suggest that a longer vaccination schedule might induce stronger and longer-lasting nAbs levels. The same principle of subunit vaccines is used for the WN-80E vaccine, which is intended to protect against WNV infection [86]. WN-80E utilizes a recombinant truncated protein comprising 80% of the N-terminal of the E protein as well, except that it is based on the WNV E protein. The same as the DENV V180 vaccine, WN-80E is produced in *Drosophila melanogaster* S2 cells, and it uses an aluminum adjuvant. A Phase I trial was performed with 25 participants who all showed high levels of nAbs; however, the durability of the nAbs was not reported.

#### 1.8.2. Virus-Like Particles

As opposed to subunit vaccines where proteins are used as antigens, VLPs take it one step further and utilize structural proteins in order to resemble a real virus. VLPs are composed of all or some of the proteins that constitute the viral capsid, which have the capability to self-assemble when recombinantly expressed. Since VLPs do not contain genomic material, there is no replication or reversion to a pathogenic state. VLPs utilize proteins arranged in dense, repetitive arrays, which are designed to induce strong cellular and humoral immune responses upon recognition. These structural properties are unique in microbial antigens, which are highly recognizable by the immune system. For this reason, VLPs have the capacity to elicit strong humoral responses since B-cells are able to specifically recognize and respond to those repetitive arrays [87]. Next to these self-adjuvating properties, VLPs offer better safety as opposed to live-attenuated and inactivated vaccines, as they are non-replicative. Several types of VLPs can be produced with several methods [64]. The difference in types can be based on the arrangement of the protein arrays, the use of an external lipid envelope or the use of chimeric VLPs. VLPs are produced in expression systems in cells, which are selected based on the requirements for protein folding and post-translational modifications. Expressions systems in bacteria, yeast, insect cells, mammalian cells and plant cells are currently used for VLP production. The difference between these systems is based on the efficiency of VLP production, assembly and maturation.

No VLP vaccine candidates for flaviviruses have yet been developed, but some studies have reported on strategies for development. Krol et al. in 2019 described two strategies that have been successful for the production of flaviviral VLPs [88,89]. Through either cis or trans expression of the M and E genes in plasmid vectors, recombinant VLPs can be formed. The inclusion of the C-protein is not a prerequisite, although it may have a stabilizing effect on VLP assembly. Another approach is the co-expression of the C, M and E structural proteins as a single cassette in conjunction with the NS2B/NS3 genes. Presently, several VLP candidates for DENV are being tested in preclinical studies, one of which has been tested in non-human primates [90].

#### 1.8.3. Viral Vector-Based Vaccines

Viral vector-based vaccines utilize unrelated and modified viruses encoding antigens designed to elicit an immune response against the target pathogen. The viral vectors enter host cells after administration, where the antigens are intracellularly expressed. As a result of the intracellular expression, both the humoral and cellular response are induced against the target pathogen. Numerous viruses have been described to be applied as platforms for vector vaccines, each with its advantages and disadvantages [91]. The difference between viral vectors can be based on which immune response they induce, which cells they enter and whether they can replicate. 

Due to the abundance of approaches for viral vector-based vaccine development, they may present potential solutions where other vaccine types have failed, especially since viral vector-based vaccines can be designed to induce both humoral and cellular immunity, which is essential against flaviviruses [92]. The viral vectors can be genetically modified to encode and produce any possible antigen, which, in turn, can go through all the post-translational modifications needed in order to increase immunogenicity. Moreover, the use of replicating vectors can potentially eliminate the need for adjuvants, as they resemble natural infections. However, viral vector-based vaccines still present some challenges and disadvantages [93]. Safety concerns have been raised for the use of replicating vectors, which may result in high viremia or persistent infections. Moreover, the viral vectors would be genetically modified organisms (GMOs), making them a potential threat to human safety if they are released into the environment [94]. In addition, these genetically modified and (non-) replicating viruses could potentially become pathogenic, making it difficult to guarantee safety [95]. Consequently, the development and production of viral vector-based vaccines are both expensive and difficult due to the complexity of designing vectors as well as safety concerns. 

As far as present flavivirus vaccine development goes, only one viral vector-based vaccine is being tested in clinical trials for ZIKV (Appendix A). The MV-ZIKV vaccine, which utilizes the measles Schwarz strain that encodes for a soluble version of the E protein for the ZIKA virus as a vector, is currently in Phase I [96]. No information on clinical safety and immunogenicity is yet available, but the vaccine has shown strong immunogenicity in preclinical studies and no fetal growth retardations in pregnant mice. 

#### 1.8.4. Epitope-Based T-/B-Cell Vaccines

Present-day bioinformatics techniques grant researchers the ability to map and predict B- and T-cell epitopes that can be used as targets for inducing specific and robust immune responses [97,98,99,100]. With the information provided by these techniques, epitope-based vaccines can be developed, which utilize epitope peptides as antigens. Generally, the design of epitope-based vaccines consists of multiple MHC-restricted epitopes, a delivery system and an adjuvant. The different types of epitope peptides include B-cell epitopes, CD8+ T-cell epitopes and CD4+ T-cell epitopes. T-cell epitopes are predominantly peptide fragments, whereas lipids, proteins, nucleic acids and carbohydrates may constitute the B-cell epitopes. These epitopes are designed to be immunodominant, which means that the immune responses are highly specific and only targeted toward the selected antigenic peptides. Following administration, these epitopes are presented on the MHC class I and II proteins of APCs. 

As with most other recombinant vaccines, epitope-based vaccines are highly safe and relatively easy to produce, but an adjuvant is needed to stimulate immunogenicity. Moreover, since the epitopes are easily degraded by host proteases, a delivery system is needed. Lei et al. 2019 report on several possibilities of delivery systems, for example, VLPs and outer membrane vesicles, with both able to function as an adjuvant as well [101].

### 1.9. Genomic Vaccines

Defined as a 3rd generation vaccine type, DNA vaccines bring new opportunities to the table for the development of vaccines against flaviviruses [102]. Whereas protein-based vaccines cause the immune system to rely on the direct administration of the antigen, DNA vaccines prompt the immune system to produce the antigen itself. The expressed protein is then able to induce both the cellular and humoral immune system at various stages. DNA vaccines contain a bacterial plasmid holding an optimized gene sequence that encodes the antigen. Through either intradermal, intramuscular or subcutaneous administration, the plasmid DNA is taken up by antigen-presenting cells (APCs) and the antigen is produced through intracellular expression. Consequently, these antigens are expressed on major histocompatibility complex (MHC) class I and II, after which the APCs travel to the draining lymph nodes. In the lymph nodes, these APCs facilitate the co-stimulation of naive CD4+ T-cells. Moreover, B-cell responses can be induced through the capturing of shed antigens by specific high-affinity B-cell receptors. In addition to the induction of the humoral response, DNA vaccines are also able to induce the cellular response [103]. Since, during the gene encoding, the antigen is intracellularly processed and expressed by the APCs, the antigen can be presented on the MHC class I protein. As a result, specific CD8+ T-cells are stimulated, which not only induces a strong cellular response but also activates and expands B- and T-cells. In addition to the ability to induce both the cellular and humoral immune system, DNA vaccines possess several advantages [104]. For example, DNA vaccines are cheap to produce compared to recombinant protein vaccines and easy to transport, making them suitable candidates for the low resource settings. Moreover, the produced antigen elicits a highly specific immune response, partly due to the antigen undergoing the same glycosylation and post-translational modifications as natural infection. Additionally, several variants of the antigen can be constructed into the plasmid antigen, allowing for immunization against a diverse set of strains. However, the use of DNA vaccines presents some safety issues as well [105]. DNA vaccines might affect gene expression through the incorporation of the plasmid DNA into the host genome. This may lead to negatively affecting cell growth or the activation of oncogenes. In addition, DNA vaccines may cause autoimmune disorders by eliciting anti-DNA antibodies. DNA vaccines are also limited in their immunogenicity and therefore need booster doses and adequate adjuvants. 

Currently, several DNA vaccine candidates are being tested in clinical trials for DENV, ZIKV and WNV, and many more in preclinical studies (Appendix A). The DNA vaccine TVDVVAX is a tetravalent DENV vaccine composed of four monovalent plasmid DNA vaccines encoding the M and E genes. TVDVVAX contains the cationic liposome adjuvant Vaxfectin, a versatile adjuvant suitable for DNA and protein-based vaccines [106]. The Phase I trial showed that TVDVVAX was well tolerated in healthy individuals and induced significant T-cell IFNy responses [76]. For ZIKV, multiple DNA vaccines are in phase trials, all using the same approach: plasmid DNA with the M and E genes [107]. These candidates all show adequate safety, although, in terms of immunogenicity, they show fewer promising results. Two DNA vaccine candidates for WNV were tested in Phase I trials. VRC302 and VRC303 both contain plasmid DNA expressing WNV M and E proteins of the NY99 strain, only differing in which promoter is utilized [108]. VRC303 was shown to be adequately safe and immunogenic, however, there has not been a follow-up clinical study since 2006. 

In addition to using plasmid DNA as a means of vaccination, mRNA can be used as well. In contrast to DNA vaccines, modified mRNA vaccines do not possess the risk of the integration of genetic material into the host genome. Similar to DNA vaccines, mRNA vaccines use the host cell machinery to produce the antigen. The mRNA contains 5′ and 3′ UTRs that ensure stability and efficient translation, as well as proprietary nucleoside modification in order to avoid induction of the innate immune system. Moderna Therapeutics developed a modified mRNA vaccine (mRNA-1325) composed of optimized mRNA encoding the ZIKV M and E structural genes [109]. Phase I trials have been completed (no results posted), and funding has been made available for Phase II and III clinical trials. Although less is known about mRNA vaccines for flaviviruses, recently, multiple mRNA vaccines have shown protection against severe SARS-CoV-2 infection globally [110]. The mRNA-1237 vaccine developed by Moderna showed 94.1% efficacy for protecting individuals from SARS-CoV-2 disease during the Phase III trial. This vaccine is composed of mRNA encoding the SARS-CoV-2 prefusion stabilized spike protein encapsulated by a lipid-based nanoparticle as delivery system. Additionally, BionTech/Pfizer developed their mRNA vaccine with a lipid-based nanoparticle as well, while utilizing nucleoside-modified RNA, which encodes the full SARS-CoV-2 spike protein [111]. Lipid-based nanoparticles are the most clinically developed delivery system, and they play a role in counteracting the lack of stability and poor efficacy that mRNA vaccines usually have. The SARS-CoV-2 pandemic showed that mRNA vaccines have several advantages over other vaccine types. Since mRNA vaccines can be developed in a cell-free process, it allows for rapid, high-scale and cost-effective production. Furthermore, mRNA vaccines can encode more than one antigen, which may be beneficial for the development of a pan-flavivirus vaccine. 

## 2. Discussion

This literature review is the first to discuss and highlight all the relevant vaccine types for the six medically important flaviviruses. We have made an inventory of all the commercially available vaccines and vaccines in clinical trials, for which we stated both the successes and challenges encountered in the development and applications. Over the last twenty years, substantial progress has been made in flavivirus vaccine development. An abundance of preclinical and clinical studies are currently being conducted for the development of flavivirus vaccines. However, efficient and safe vaccines for DENV, ZIKV and WNV have yet to reach the market, as several barriers are still to be overcome. Moreover, some of the presently available vaccines for YFV, JEV and TBEV still have room for improvement. Based on the available literature and epidemiological data, a mass vaccination program should be considered for some flavivirus infection, e.g., dengue and possibly Zika. However, the major point of concern is even distribution and availability of vaccines to low-income countries, as flavivirus-endemic areas are located mostly in tropical and subtropical regions, which lack well-equipped hospitals and trained personnel. In general, the biopharmaceutical industry should focus more on those that need a vaccine most, rather than who can afford it. Mostly, multiple doses of vaccine are required to induce protective immunity against the virus, thus it is difficult for individuals with a low economic condition to afford multiple doses of an expensive vaccine. Therefore, the characteristics of a good vaccine should not be limited to its safety and efficacy, but it should be inexpensive and widely available throughout the globe [112]. The goal of this review was to evaluate the several relevant vaccine types and vaccines for flaviviruses, as well as to gain insights into future strategies for the development of flavivirus vaccines. Presently, live-attenuated and inactivated vaccines account for all the commercially available vaccines for flaviviruses [54,56]. However, several issues present themselves in both commercially available and phase-trial live-attenuated and inactivated vaccines. 

Live-attenuated vaccines can be advantageous, as one dose could induce lifelong immunity, which does require a potentially harmful level of viremia to be established. The potential of live-attenuated vaccines to induce both strong humoral and cellular immune responses may provide solutions where other vaccine types cannot. For live-attenuated vaccines, the most frequent concerns involve suboptimal safety. For example, severe dengue has been observed in patients who were vaccinated prior to secondary infection with the live-attenuated vaccine Dengvaxia [63,64]. Several studies hypothesize that this is due to ADE since this phenomenon is increasingly observed in patients who were seronegative upon vaccination. Moreover, Castanha et al. 2017 report that the presence of anti-DENV antibodies can enhance ZIKV infection through ADE as well [113]. In addition, several rare serious adverse events have been described regarding the YFV 17D vaccines, such as neurologic syndromes, hypersensitivity, GBS and severe viscerotropic disease (YEL-AVD). Little is known about the disease mechanism, although YEL-AVD is characterized by high levels of replication by the 17D virus, similar to those of natural infections [114]. 

Several studies report on approaches to reduce toxicity from live-attenuated vaccines. For the ChimeriVax-WN02 vaccine, three mutations were inserted on the E protein, thereby reducing neurovirulence [115]. Kwek et al. 2018 describe an approach for a live-attenuated vaccine, where a ZIKV variant produces small plaques due to interferon-restricted propagation [116]. This resulted in a reduction of viremia in the vital organs of mice, while still inducing strong and sufficient immunity. This approach may also be applicable to other flaviviruses. While live-attenuated vaccines bring various safety concerns, they might still offer solutions to flavivirus vaccines. At present, several live-attenuated vaccines are in phase trials for DENV and WNV, most of which show promising immunogenicity and tolerance.

Some of the safety concerns in live-attenuated vaccines could be diminished if non-live vaccines, for example, inactivated vaccines, would be used. Since inactivated vaccines induce a weaker cellular response compared to live-attenuated vaccines, they might overcome the ADE challenge in DENV vaccines. Furthermore, since inactivated vaccines do not allow for viral replication, the risk for YEL-AVD could be diminished as well [114]. Consequently, an inactivated vaccine for YFV is currently in phase trials, with the goal to increase safety. Although inactivated vaccines may be safer than live-attenuated, it may be difficult to retain the same immunogenicity as live-attenuated vaccines. Several strategies are described in the literature to potentially improve immunogenicity, such as alternatives to formalin for inactivation and the use of various adjuvants. The inactivation techniques are crucial in the success of inactivated vaccines since inadequate inactivation could lead to the administration of infectious particles or the damaging of the relevant epitopes, resulting in poor immunogenicity. The inactivated vaccines currently in phase trials for DENV, ZIKV, WNV and YFV have not yet passed the Phase II trials, as some struggle with maintaining sufficient immunogenicity, especially for ZIKV. Several inactivation strategies and adjuvants are being evaluated for the improvement of immunogenicity. For example, Fernandez et al. 2015 described the use of adjuvant systems for the TDENV-PIV vaccine, which significantly increased immunogenicity while maintaining safety [78]. Moreover, the same vaccine is inactivated by psoralen, which showed a 30–60% increased binding capacity of mAbs compared to inactivation with formalin or azide [117].

Recombinant vaccines provide ways to minimize safety concerns as well, as there is no need to include potentially toxic or unnecessary components [118]. Recombinant vaccines are designed to induce highly characterized immune responses with only specific parts of a pathogen. With this characteristic, they might provide safe and efficient vaccines for viral pathogens where traditional approaches have failed. For example, successful subunit vaccines have already been developed for hepatitis B and human papillomavirus [119]. Two subunit vaccines for DENV and WNV have been tested in Phase I trials, however, these trials resulted in insufficient data for immunogenicity [85]. Although subunit vaccines for flaviviruses are far from becoming commercially available, numerous preclinical studies are being conducted with promising candidates. The use of bacterial flagellin is employed in various preclinical approaches for DENV, ZIKV and WNV, as the flagellin is able to trigger the toll-like receptor 5 (TLR5) response [120,121,122]. This approach has shown success in vaccine development for influenza, as it links the humoral and innate immunity, which makes the vaccine resemble a natural infection [123,124]. Other approaches for DENV, ZIKV and WNV describe various ways of producing a recombinant E protein, which is the major antigen for generating nAbs [88]. Many of them report high immunogenicity and safety in preclinical studies but have yet to be tested in clinical trials. 

Another recombinant approach for flavivirus vaccines could be the development of VLPs [64]. Since VLPs resemble infectious viral particles while also presenting highly recognizable antigens, VLPs often elicit stronger immune responses than subunit vaccines. VLPs bring various advantages to the table, such as low costs, easy development and high safety, although little is known about clinical use. Several preclinical studies on VLP vaccine candidates are currently being conducted for DENV and ZIKV, some of which have been tested in NHPs [125]. Presently, there is a plethora of different approaches to develop VLPs, as there are over 170 different expression systems available for use. VLPs have also been shown to effectively stimulate both CD4+ and CD8+ T-cell responses against the hepatitis C virus, an essential part of the immune response against flaviviruses [30,126]. While VLPs provide promising properties for the development of flavivirus vaccines, there is still much left undiscovered. Future research will determine whether flavivirus VLPs are suitable for clinical use. 

Viral vector-based vaccines offer similar beneficial characteristics for flavivirus vaccine development. The use of a viral vector provides the ability to induce strong immunogenicity and resemble natural infections, as well as a multitude of approaches. On the other hand, it necessitates the use of a GMO, and a probability exists that the viral vector may become pathogenic and even transmit to other people [95]. A Phase I trial has been conducted on the measles-vectored MV-ZIKV vaccine, which showed promising results in preclinical studies but has yet to report clinical data [96]. Viral vector-based vaccines can potentially induce highly potent immune responses required for some flaviviruses [93]. However, the complexity of both the development and immune mechanism still provides barriers to development.

Additionally, the promising but fairly unexplored field of epitope-based vaccines offers opportunities for flaviviruses as well. As mentioned before, CD4+ and CD8+ T-cells are crucial in the immune response against flaviviruses, which can be specifically induced with epitope-based vaccines [29]. Meanwhile, several vaccine types lack the ability to induce sufficient T-cell responses. Cunha-Neto et al. 2017 describe a design for a CD8+ T-cell-inducing vaccine for ZIKV by using previously identified DENV class I epitopes, as well as class II epitopes for CD4+ T-cells, which shows promising preclinical data.

Alternatively, DNA and mRNA vaccines can also induce both humoral and cellular immune responses [102]. These genomic vaccine types are relatively safe and easy to produce and can be administered in various ways. However, for the facilitation of an immune response, the DNA or mRNA needs entry to the cytoplasm of APCs prior to antigen presentation. For this reason, achieving sufficient immunogenicity in humans has proven to be difficult. Currently, genomic vaccines for flaviviruses are mostly in the early stages of development, several of which struggle with inducing adequate immunogenicity. However, the DNA vaccines TVDV-VAX for DENV and VRC5283 for ZIKV both have shown promising immunogenicity in Phase I/II trials [106,107]. Several studies have tested and explored strategies that may improve the immunogenicity of genomic vaccines. This includes the use of adjuvant and delivery systems, promotor selection, the use of different administration strategies, antigen codon optimization, electroporation and several others [127,128,129]. For example, Farris et al. 2016 reported an approach to use microparticulate as a delivery system, which may enhance vaccine uptake in APCs [130]. The progress made in the last two decades in improving DNA vaccine immunogenicity currently provides versatile approaches for flavivirus DNA vaccine development.

## 3. Conclusions

In conclusion, different vaccine types offer different efficacies, and there is often a trade-off between immunogenicity and efficacy. DENV, WNV and especially ZIKV require potent immune responses, which some vaccine types struggle to achieve. Considerable progress has been made on improving immunogenicity and reducing safety concerns of the different vaccine types, which makes way for versatile approaches for flavivirus vaccine development. Moreover, the development of more than one vaccine type for a particular flavivirus could be useful. For example, for ZIKV, a live-attenuated vaccine could be used for healthy people, whereas an inactivated or recombinant vaccine could be used for pregnant women and infants. Interestingly, CYD-TDV, a tetravalent vaccine based on the yellow fever virus backbone, has been recommended for use in children with evidence of past dengue infection. However, TAK-003, a live-attenuated vaccine showed protection among children with or without a history of dengue infection [67]. To this day, vaccination-mediated prevention remains the most suitable approach for combating flaviviruses. Although, additional studies are needed to pass the hurdles currently facing the vaccine development for DENV.

## Figures and Tables

**Figure 1 viruses-15-00860-f001:**
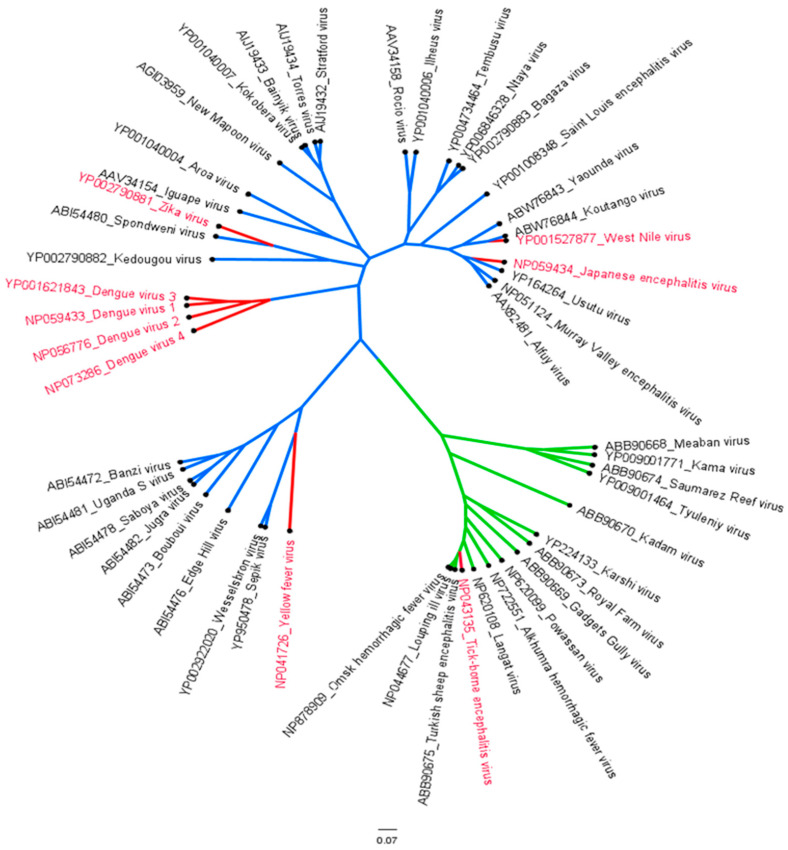
Bayesian evolutionary analysis sampling tree cladogram of the mosquito- (blue) and tick- (green) borne flavivirus-infecting vertebrates. The tree was constructed using the Bayesian Markov chain Monte Carlo method available in MrBayes v.3.2.3 [3]. The six medically relevant viruses, DENV (serotype 1–4), ZIKV, WNV, YFV, JEV and TBEV, have been highlighted in red.

**Figure 2 viruses-15-00860-f002:**
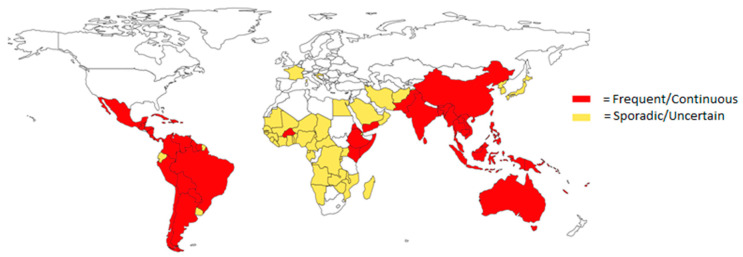
Countries at risk for DENV infections. Countries labeled Frequent/Continuous are currently experiencing DENV outbreaks or ongoing DENV transmission, while Sporadic/Uncertain suggests a varying and unpredictable risk and the unavailability of country-wide data [11]. Map created using QGIS software (small countries not included).

**Figure 3 viruses-15-00860-f003:**
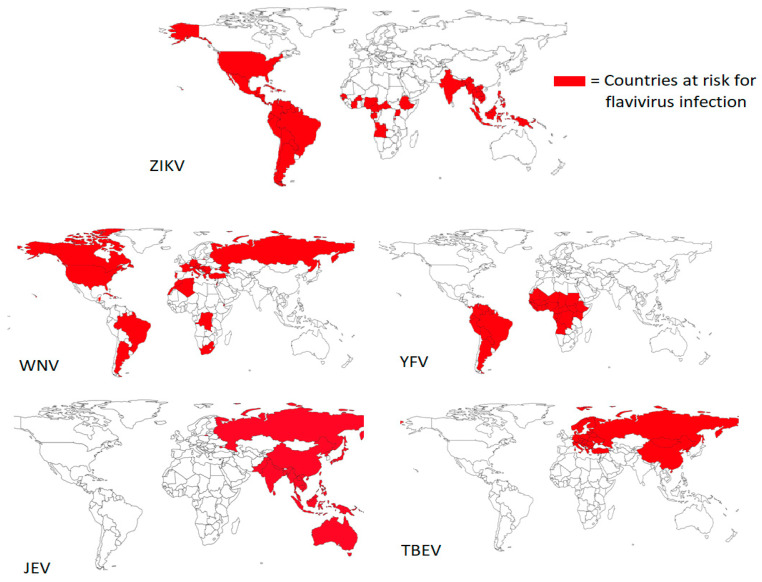
The global distribution and countries at risk for flavivirus infections. ZIKV is found throughout the sub-tropical and tropical regions of the Americas, Africa and Asia [9]. The risk for WNV infection is localized in Europe, parts of Africa and the Americas [12,13], whereas the risk for YFV is only found in South America and Africa [14]. Ongoing transmission of JEV occurs in Asia and Oceania [15,16], and TBEV is distributed throughout Europe and northeast Asia [17,18]. Map created using QGIS software (small countries not included).

**Figure 4 viruses-15-00860-f004:**
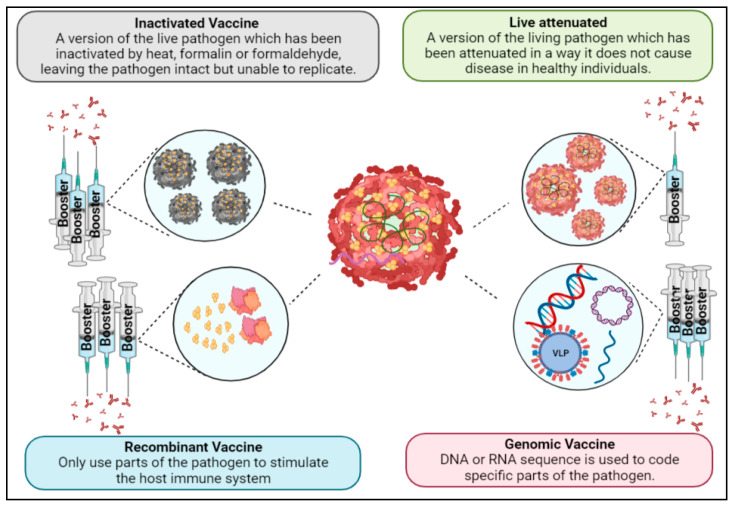
Flavivirus vaccine strategies.

**Table 1 viruses-15-00860-t001:** Symptoms, complications, geographical distributions and vectors of flaviviruses as described by the CDC.

Flavivirus	Mild Symptoms	Severe Symptoms	Complications	Regions	Vectors
DENV	Fever, nausea, vomiting	Hemorrhagic fever, hematemesis	Cardiomyopathy, shock syndrome	The Americas, Africa, the Middle East and the Pacific Island	*Ae. aegypti, Ae. albopictus*
ZIKV	Fever, rash, headache	Long-term neurological complications	Birth defect, in rare cases GBS or cerebral edema	The Americas, Africa, Asia and the Pacific Island	*Ae. aegypti, Ae. albopictus*
WNV	Fever, headache, body aches	High fever, coma, tremors, convulsions, vision loss, numbness and paralysis, myalgia	Encephalitis, meningitis	Africa, Europe the Middle East, North America and west Asia	*Culex pipiens*
YFV	Fever, chills, severe headache	High fever, jaundice, bleeding	Shock, organ failure	Africa, central and south America	*Aedes aegypti, Aedes africanus, Haemagogus spp, Sabethes spp.*
TBEV	Fever, malaise, anorexia, muscle aches, headache, nausea and/or vomiting	Drowsiness, confusion, sensory disturbances, paralysis	Encephalitis, meningitis, meningoencephalitis	Europe and Asia	*Ixodidae ricinus*
JEV	Fever, headache, vomiting	Neurologic symptoms, seizures	Encephalitis, neurologic, cognitive or psychiatric symptoms after disease	Southeast Asia and the Pacific Island	*Culex tritaeniorhynchus*

**Table 2 viruses-15-00860-t002:** List of FDA-approved vaccines for flaviviruses.

Name	Vaccine Type	Virus	Manufacturer
CYD-TVD/Dengvaxia	Chimeric live-attenuated	DENV	Sanofi Pasteur
IC51/IXIARO	Inactive	JEV	WRAIR
JE-VAX	Inactive	JEV	The Research Foundation for Microbial Disease of Osaka University
SA 14-14-2	Live-attenuated	JEV	Chengdu Institute of Biological Product
IMOJEV/JE-CV	Live-attenuated	JEV	Sanofi Pasture
TBE-Moscow	Inactive	TBEV	Chumakov Institute of Poliomyelitis and Viral Encephalitides
EnceVir	Inactive	TBEV	Microgen
FSME-IMMUN	Inactive	TBEV	Baxter
Encepur	Inactive	TBEV	Novartis
YFV-17DD	Live-attenuated	YFV	Bio-Manguinhos (Fiocruz)
YFV-17D-204	Live-attenuated	YFV	Sanofi Pasteur Institute Chiron/Novartis
YFV-17D-213	Live-attenuated	YFV	Federal State Unitary Enterprise of Chumakov Institute

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
