# Peer review of "A Perspective on Current Flavivirus Vaccine Development: A Brief Review"

_viruses, 2023, doi:10.3390/v15040860_

Round 1

Reviewer 1 Report

This is an improved version of manuscript that reviews the major types of vaccine in flavivirues, especially including the status of preclinical/clinical trials for the six medically important flaviviruses. In this review the author concluded a comprehensive view of different type of vaccines used in flaviviruses study including cons and pros as well as challenges. Overall, this is a very useful review that can provide some insights and directions of vaccine development for both academy and industry. However, there are  still some minor parts should be revised:

1. In the section 1.3 Immune response, E, M and NS1-induced Abs were all mentioned on line 147, but only E and M-induced Abs were further described and discussed. It would be nice to address more about the immune response of NS1-induced antibodies. The pathogenesis of flavivirus NS1 were widely studied and proved in some clinical trials. Either protective and pathogenic roles of anti-NS1 were also studied and published.

2. In the section 1.8.4 Epitope-based T/B cell vaccines, the author only described the rationale and mechanism of this vaccine type with adjuvant-related study. Although there are no Epitop-based T/B cell vaccines for flaviviruses in clinical trials yet,  it would be nice to cite some references of in vivo study of flaviviruses.

3. The references shown in text somehow did not match  the ones in the Reference section. Most of the ref. in the text match the next one in the Reference section. (Ex. the ref. 71 in the text should match the ref. 72 in the Reference list.) Please check the consistency.

Author Response

Response to Reviewers’ comments

Reviewer 1:

Comments and Suggestions for Authors

  1. In the section 1.3 Immune response, E, M and NS1-induced Abs were all mentioned on line 147, but only E and M-induced Abs were further described and discussed. It would be nice to address more about the immune response of NS1-induced antibodies. The pathogenesis of flavivirus NS1 were widely studied and proved in some clinical trials. Either protective and pathogenic roles of anti-NS1 were also studied and published.

Response: We are very grateful to the reviewer for his/her constructive comments. As suggested, the authors have added the role of NS1 induced antibodies response in flavivirus vaccine development in section 1.3. Immune response line 182 -188. 

  1. In the section 1.8.4 Epitope-based T/B cell vaccines, the author only described the rationale and mechanism of this vaccine type with adjuvant-related study. Although there are no Epitop-based T/B cell vaccines for flaviviruses in clinical trials yet, it would be nice to cite some references of in vivo studies of flaviviruses.

Response: Thank you for your valuable comment. In line with reviewer comments. As suggested by the reviewer three references (96 -99) related to in vivo study of flaviviruses have been added in section 1.8.4 Epitope-based T/B cell vaccines.

  1. The references shown in text somehow did not match the ones in the Reference section. Most of the ref. in the text match the next one in the Reference section. (Ex. the ref. 71 in the text should match the ref. 72 in the Reference list.) Please check the consistency.

Response: Thank you for your valuable comment. The consistency of all references in the text has been thoroughly checked and matched with the Reference section.

Reviewer 2 Report

Overall, this is a fine review, but it lacks references to support much of the information throughout many of the sections prior to the discussion section. Discussion section fails to discuss ethical consideration (such as accessibility, affordability, reasonable expectations for immunity (if multiple boosters are required for protection, will they be easily accessible beyond the first dose?). These conversations need to be front and center when it comes to vaccine development. The author only mentions pregnant women and children in the conclusion in the context of ZIKV, but fails to acknowledge the prominence of pediatric DENV infections. While this review is acceptable as is, I think it is not advisable to publish this work without the addition of important ethical considerations.

This review would really benefit from an infographic summarizing the types of vaccine strategies.

Line Edits/Suggestions:

- Line 31: Zika should be capitalized always, since it's named for a geographical area.

- Line 32: no hyphen between West Nile.

- Table 1: this table is difficult to read the way it is currently formatted. I suggest adding horizontal lines between each row, or alternating shading of rows.

- Table 1: DENV vector Ae. aegypti and Ae. albopictus should be italicized. Also throughout the vector column, check for inappropriate capitalization (species names).

- Line 45: replace period (.) with comma (,) in 20,000, and all large numbers going forward (see line 94, etc.).

- Line 52: need reference for ADE mention. Look at the work from Eva Harris' lab for good references on this.

- Line 78-85: this paragraph fails to mention the resurgence of YFV in South America due to generational gaps in vaccine efforts. Seems like a major oversight given the focus of this review.

- Line 93: spellcheck "neregionc". Did you mean "neurologic"?

- Line 110: Avoid terms like "and others" in a review. Given the overlapping symptomatology of many of these infections, it is really important to discuss all discrete symptoms and provide information on how to obtain differential diagnoses.

- Line 211/section 1.5: This section fails to provide adequate references.

- Line 256/section 1.6: This section fails to provide adequate references.

- Line 374: sentence starting with "However..." uses past tense ("was" instead of "is").  Careful with tenses.

- Lines 392-400: references?

- Line 409: Sentence starting with "However.." regarding ZIKV. Are you suggesting the nAb levels resulting from vaccination against DENV are not high enough to also protect against ZIKV? Rephrase for clarity.

- Line 510: Claim of GMOs being a "potential threat to human safety" is vague. Explain and back up with references, or remove this sentence in favor of the one immediately following which is more specific.

- Line 559: consider using "low-resource settings" rather than "developing world".

- Line 566: Do you have evidence of anti-DNA antibodies and the mounting of an autoimmune response or impact on oncogenes as a result of genomic vaccines? Please reference these claims accordingly.

- Line 588: add hyphen to SARS-CoV-2.

Author Response

Response to Reviewers’ comments

Reviewer 3:

Comments and Suggestions for Authors

  1. Overall, this is a fine review, but it lacks references to support much of the information throughout many of the sections prior to the discussion section. Discussion section fails to discuss ethical consideration (such as accessibility, affordability, reasonable expectations for immunity (if multiple boosters are required for protection, will they be easily accessible beyond the first dose?). These conversations need to be front and center when it comes to vaccine development. The author only mentions pregnant women and children in the conclusion in the context of ZIKV but fails to acknowledge the prominence of pediatric DENV infections. While this review is acceptable as is, I think it is not advisable to publish this work without the addition of important ethical considerations.

Response: Thank you for your valuable comments. Based on the reviewer’s comment ethical considerations have been discussed in the discussion section (lines 672-682).

Based on the reviewer’s comments “Interestingly, CYD-TDV, a tetravalent vaccine based on yellow fever virus backbone, have been recommended to use for children with evidence of past dengue infection. However, TAK-003, a live attenuated vaccine showed protection among children with or without a history of dengue infection [66].” (lines 808-811) statements have been incorporated in the conclusion section to acknowledge the prominence of pediatric DENV infection.

  1. This review would really benefit from an infographic summarizing the types of vaccine strategies.

Response: We are very grateful to the reviewer for his/her constructive comments. Based on the reviewer’s comment an infographic summary of different types of vaccines has been added (Figure 4.).

  1. Line 31: Zika should be capitalized always since it's named for a geographical area.

Response: Thank you for your valuable comment. In line with the reviewer’s comments, Zika has been capitalized throughout the review.

  1. Line 32: no hyphen between West Nile.

Response: Thank you for your valuable comment. In line with reviewer comments. Hyphen has been removed between West Nile.

  1. Table 1: this table is difficult to read the way it is currently formatted. I suggest adding horizontal lines between each row, or alternating shading of rows.

Response: Thank you for your valuable comment. In line with the reviewer’s comment, the style of “Table 1” has been formatted.

  1. Table 1: DENV vector Ae. aegypti and Ae. albopictus should be italicized. Also, throughout the vector column, check for inappropriate capitalization (species names).

Response: Thank you for your valuable comment. In line with the reviewer’s comment, the vector names have been formatted throughout the review article.

  1. Line 45: replace period (.) with comma (,) in 20,000, and all large numbers going forward (see line 94, etc.).

Response: Thank you for your valuable comment. In line with the reviewer’s comment, all the large numbers have been formatted according to the reviewer’s comment throughout the article.

  1. Line 52: need reference for ADE mention. Look at the work from Eva Harris' lab for good references on this.

Response: Thank you for your valuable comment. In line with the reviewer’s comment, a reference (6) has been added to line 54 (previously line 52).

  1. Line 78-85: this paragraph fails to mention the resurgence of YFV in South America due to generational gaps in vaccine efforts. Seems like a major oversight given the focus of this review.

Response: Thanks for your valuable comment. In the line with the reviewer’s comment, the resurgence of YFV in South America has been added to the second paragraph (lines 88-95) of section 1.1 Epidemiology of flaviviruses to discuss.

  1. Line 93: spellcheck "neregionc". Did you mean "neurologic"?

Response: Thanks for your valuable comment. In line with the reviewer’s comment, the word “neregionc” in line 106 (previously line 93) has been replaced with "neurologic".

  1. Line 110: Avoid terms like "and others" in a review. Given the overlapping symptomatology of many of these infections, it is really important to discuss all discrete symptoms and provide information on how to obtain differential diagnoses.

Response: Thanks for your valuable comment. In line with the reviewer’s comment, all discrete symptoms of TBEV have been discussed in the text line 123 previously line 110. 

  1. Line 211/section 1.5: This section fails to provide adequate references.

Response: Thanks for your valuable comment. Adequate references have been added to section 1.5 Vaccine development.

  1. Line 256/section 1.6: This section fails to provide adequate references.

Response: Thanks for your valuable comment. Adequate references have been added to section 1.6 Live attenuated vaccines.

  1. Line 374: sentence starting with "However..." uses past tense ("was" instead of "is").  Careful with tenses.

Response: Thanks for your valuable comment. In line with the reviewer’s comment, the tense has been corrected in "However..." line 374. 

  1. Lines 392-400: references?

Response: Thanks for your valuable comment. In line with the reviewer’s comment, a reference has been added to lines 392-400.

  1. Line 409: Sentence starting with "However.." regarding ZIKV. Are you suggesting the nAb levels resulting from vaccination against DENV are not high enough to also protect against ZIKV? Rephrase for clarity.

Response: Thanks for your valuable comment. In line with the reviewer’s comment, the sentence “However, it seems that for protective immunity against ZIKV, there is a need for higher nAbs levels.” Rephrased and explained to “Multiple purified inactivated ZIKV vaccine candidates have been entered into phase-I trials, out of which four have completed the phase-I trial. Based on available data, two out of four vaccines, namely ZPIV (purified, formalin-inactivated Zika vaccine) and PIZV (purified, formalin-inactivated, alum-adjuvanted whole Zika virus vaccine candidate) were reported to be safe, well tolerated and induced substantial amounts of antibodies in healthy individuals [78-80]”. (Lines 452-458 previously 409)

  1. Line 510: Claim of GMOs being a "potential threat to human safety" is vague. Explain and back up with references, or remove this sentence in favor of the one immediately following which is more specific.

Response: Thanks for your valuable comment. In line with the reviewer’s comment, the sentence (line 562 previously line 510) “Moreover, the viral vector would be genetically modified organisms (GMOs), making them a potential threat to human safety” has been rewritten to “Moreover, the viral vector would be genetically modified organisms (GMOs), making them a potential threat to human safety, if they are released into the environment [93].” Followed by backing up with a reference “93. Baldo A, van den Akker E, Bergmans HE, Lim F, Pauwels K. General considerations on the biosafety of vi-rus-derived vectors used in gene therapy and vaccination. Curr Gene Ther. 2013 Dec;13(6):385-94”.

  1. Line 559: consider using "low-resource settings" rather than "developing world".

Response: Thanks for your valuable comment. In line with the reviewer’s comment, the "low-resource settings" is replaced with "developing world", in line 612 previously 559.

  1. Line 566: Do you have evidence of anti-DNA antibodies and the mounting of an autoimmune response or impact on oncogenes as a result of genomic vaccines? Please reference these claims accordingly.

Response: Thanks for your valuable comment. In line with the reviewer’s comment, reference 104 has been added to justify the claims.

  1. Line 588: add hyphen to SARS-CoV-2.

Response: Thanks for your valuable comment. In line with the reviewer’s comment, “SARSCoV-2” has been changed to “SARS-CoV-2”.